# The Estimation of Geothermal Reservoir Temperature Based on Integrated Multicomponent Geothermometry: A Case Study in the Jizhong Depression, North China Plain

**Junzu Deng [1], Wenjing Lin [2,3,*], Linxiao Xing [2,3] and Li Chen [2,3]**

1   Hydrogeology and Engineering Geology Survey Institute of Guangxi, Liuzhou 545006, China
2   The Institute of Hydrogeology and Environmental Geology, Chinese Academy of Geological Sciences, Shijiazhuang 050061, China
3   Technical Innovation Center for Geothermal and HDR Exploration and Development, Ministry of Natural Resources, Shijiazhuang 050061, China
*   Correspondence: linwenjing@mail.cgs.gov.cn

**Abstract:** The coal-dominated energy structure in the Beijing–Tianjin–Hebei region has caused serious air pollution and contradicts the construction of a clean, low-carbon, safe and efficient energy system. Substituting geothermal energy for fossil energy such as coal can effectively alleviate this problem. Located in the hinterland of the Beijing-Tianjin-Hebei region, the Jizhong Depression is rich in geothermal resources and has great development potential, though the degree of current development and utilization is not high. Vigorously developing geothermal energy can not only effectively alleviate the air pollution problem in the Beijing–Tianjin–Hebei region, but also optimize the regional energy structure. Geothermal reservoir temperature determines the development and utilization value of geothermal resources, and accurate evaluation of the geothermal reservoir temperature of geothermal resources can provide a reliable basis for the subsequent development of geothermal resources in the Jizhong Depression. Aiming at the commonly used sandstone geothermal reservoir and carbonate geothermal reservoir in the Jizhong Depression, this paper collected 24 sandstone geothermal reservoir geothermal fluids and 14 carbonate geothermal reservoir geothermal fluids in the central-southern area of the Jizhong Depression and a water chemistry test was carried out. According to the test results of water chemistry, the temperature of the geothermal reservoir is estimated by using the cation geothermometer, the $SiO_2$ geothermometer and the multi-mineral equilibrium method, and it is compared with the actual temperature measurement results of the boreholes. The results show that the direct use of a geothermal geothermometer for calculation will cause large errors. Through water–rock balance analysis, the use of a Na-K-Mg balance diagram, $SiO_2$ and $1000/T$ relationship diagram and Na/K and $1000/T$ relationship diagram can determine whether the geothermal fluid is suitable for the geothermometer, which can effectively reduce the error. The chalcedony geothermometer in the central and southern part of the Jizhong Depression is the most suitable. The multi-mineral balance method, the Na-K geothermometer and the K-Mg geothermometer have also achieved good results, while the quartz and Na-K-Ca geothermometers are not suitable for the south-central Jizhong Depression area.

**Keywords:** geothermometer; geothermal reservoir temperature; Jizhong Depression; geothermal resources; conductive geothermal systems





## 1. Introduction

In recent years, the coal-dominated energy structure in the Beijing–Tianjin–Hebei region and the environmental pollution caused by industrial transfer have made air pollution in the Beijing–Tianjin–Hebei region increasingly intensified, with heavy haze weather occurring frequently [1]. In order to win the "Blue Sky Defense War", the government

has reduced the use of bulk coal by increasing the proportion of central heating and reducing the use of coal in vast rural areas through "coal to gas" and "coal to electricity" policies. However, at the same time, the Beijing–Tianjin–Hebei region has also experienced an unusually serious "gas shortage" and the natural gas supply rate has entered an orange warning, which means that the energy supply and demand gap has reached 10% to 20% and that energy structure contradictions have become increasingly prominent [2]. Therefore, vigorously developing clean and environmentally friendly energy sources to replace fossil energy sources, such as coal and petroleum, is the only way to effectively solve air pollution [3].

Geothermal energy is a green energy that integrates "heat, mineral, and water". Compared to other energy sources, it has the advantages of low cost, sustainable use and environmental protection [4]. Located in the hinterland of the Beijing–Tianjin–Hebei region, the Jizhong Depression is rich in geothermal resources, has hot water resources that are widely distributed, and has large development potential is [5], though the degree of current development and utilization is not high. The vigorous development of geothermal energy can effectively alleviate the Beijing–Tianjin–Hebei air pollution problem and contribute to the optimization of the energy structure of the Beijing–Tianjin–Hebei region while keeping the achievement of carbon peak and carbon neutral targets on schedule.

In the research, development and utilization of hydrothermal geothermal resources, geothermal reservoir temperature is an important parameter for dividing the genesis types of geothermal systems and evaluating the potential of geothermal resources [6]. It determines the development and utilization value of geothermal resources. Accurately judging the temperature of thermal storage is one of the most important tasks in the evaluation of geothermal resources. There are two main methods for determining the geothermal reservoir temperature: a ground temperature measurement method and a calculation method. The ground temperature measurement method is accurate and reliable but is time-consuming and expensive, and it is usually difficult to measure directly [7]. Therefore, calculation methods are often used to determine the geothermal reservoir temperature, such as the geothermometer method; however, the temperature estimated by the geothermometer method often has a large error in relation to the actual temperature, and the results of the geothermometer calculations are quite different. In order to determine the practicability of the geothermal temperature scale in the Jizhong Depression, this study will provide a scientific basis for the subsequent development of geothermal resources. This article estimates the geothermal reservoir temperature based on multiple components such as the cationic geothermometer, the silica geothermometer and the multi-mineral balance method. The estimated value of the geothermometer is compared to the measured temperature value, the actual effect of each geothermometer is evaluated, and the most suitable geothermometer in the Jizhong Depression is selected to provide a basis for the subsequent development and utilization of geothermal resources.

## 2. Geological Setting

The study area is located in the Jizhong Depression in the abdomen of the Beijing–Tianjin–Hebei region (see Figure 1a). The area is covered by Quaternary sediments with a thickness of about 260–500 m, loose structure, large porosity and poor thermal conductivity [8]. It has a good thermal insulation and water barrier effect and is an ideal geothermal reservoir cap layer. There are multiple layers of high-porosity, high-permeability sandstone thermal reservoirs under the sedimentary layer, and the base rock area below has a carbonate thermal reservoir with higher thermal conductivity, which is a commonly used thermal reservoir in the region [9]. The area has undergone multiple periods of tectonic movement, forming a concave–convex structure from west to east which can be divided into 12 depressions and 7 uplifts [10] (see Figure 1b), including the Beijing Depression, Daxing Uplift, Langgu Depression, Niutou Uplift, Baxian Depression, etc. The recessed area and the raised area form a fracture, forming a good water and heat conduction channel. The area is located in the upper mantle uplift area, and the heat source is relatively shallow.

One feature is upper mantle heat and radioactive heat in the bedrock granite [11]. The heat is accumulated in the form of heat conduction which is then the mainstay, while convection is the auxiliary to transfer heat to the surface [12]. From the perspective of "covering, storage, channel, and source", the geological conditions in the Jizhong Depression are conducive to the formation of geothermal resources. There are also many geothermal fields in the area, such as the Niutuo geothermal field and the Rongcheng geothermal field.

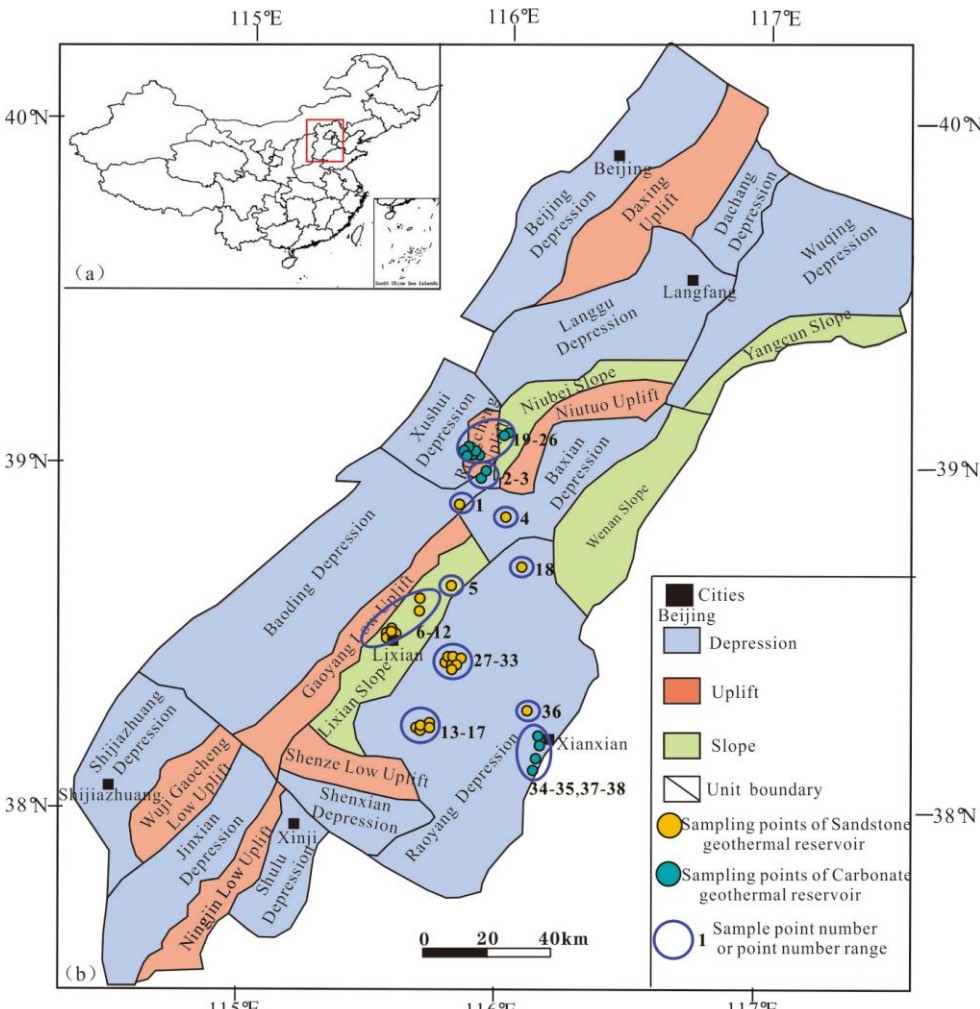

**Figure 1.** (**a**) Location of the study area in China (red box shows the area where the study area is located). (**b**) Tectonic zoning map of the Jizhong Depression (modified from Mao et al. [13], 2021).

## 3. Material and Methods

This research mainly includes: ① Sampling and testing of geothermal fluid samples. ② Geothermal well logging (mainly through data collection). ③ Using geothermal geothermometers and the SI method to estimate geothermal reservoir temperature. ④ Estimated results compared with the actual logging temperature which are discussed and then summarized.

### 3.1. Geothermal Fluid Samples

Focusing on the sandstone geothermal reservoir and carbonate geothermal reservoir commonly used in the Jizhong Depression, 38 sets of geothermal fluid samples (all of which are geothermal boreholes) were collected from multiple geothermal fields in the central and southern part of the Jizhong Depression. There are 24 sets of sandstone geothermal reservoir geothermal fluids and 14 sets of carbonate geothermal reservoir geothermal fluids.

All geothermal wells are completed heating wells, and the sampling of geothermal wells is mainly concentrated in the winter heating period. The sampling layer is mainly located in the middle of the thermal reservoir utilization section, and sampling depth ranges from 900 to 2190 m. The sampling points are shown in Figure 1b. The geothermal fluid samples were sent to the Key Laboratory of Groundwater Science and Engineering of the Ministry of Land and Resources for full water quality analysis. Detection was conducted in accordance with the National Standards of the People's Republic of China for Drinking Natural Mineral Water Inspection Method (GB/T8538-2008), in which cations are used for ion concentration detection by ICP and anions are used for ion chromatography. The test results are shown in Table 1 and the electric neutrality principle is used to test the accuracy of water quality analysis. According to the requirements of water quality analysis, the charge balance error E of the water sample in a balanced state should be less than 5%. The balance error of this geothermal water sample is between $-1.5\%$ and $3.9\%$. Within the allowable error range, water chemical analysis can be performed. $SiO_2$ in groundwater often exists in the form of silicic acid, so it can be converted into $SiO_2$ by testing the content of silicic acid in the water.

### 3.2. Geothermal Logging

When drilling to expose and penetrate the thermal reservoir, use a thermistor or other well temperature meter to directly measure the temperature in the well and then use the average temperature of the top plate and bottom plate of the thermal reservoir as the geothermal reservoir temperature. Geothermal logging mainly uses ShenKai series logging tools to perform temperature measurement 72 h after the completion of the well, and the obtained logging data are near-steady-state temperature data. According to our long-term monitoring of geothermal wells, the difference between the results measured after 72 h and the results after half a year is less than 2 °C. It can therefore be used as the real temperature.

### 3.3. Geothermometers

The geothermal geothermometer refers to the use of the hot water chemical component concentration related to the temperature of the underground geothermal reservoir or the relationship between its ratio and temperature to estimate the temperature of the deep geothermal reservoir. The principle is that the deep geothermal reservoir reaches a water–rock balance and that the hot water is rising. The temperature drops, but the chemical composition remains relatively stable, which can be used to estimate the temperature of the deep geothermal reservoir. The application of a geochemical geothermometer has the following conditions: ① The reaction in deep hot water is only related to temperature. ② The components involved in the temperature-dependent reaction must have sufficient abundance. ③ The deep hot water reaches the water–rock balance. ④ When the hot water flows to the surface under low temperature conditions, the components do not change or change very little. ⑤ The hot water is not mixed with cold water on the way up [14]. Various quantitative geothermal geothermometers are usually used to estimate the temperature of deep geothermal reservoirs. The most commonly used geothermal geothermometers include the cation geothermometer and $SiO_2$ geothermometer.

Table 1. Table of major ion concentrations in geothermal fluids (mg/L).

| NO. | Reservoir Lithology | $Ca^{2+}$ | $Mg^{2+}$ | $Na^+$ | $K^+$ | $Cl^-$ | $HCO_3^-$ | $SO_4^-$ | $SiO_2$ |
|---|---|---|---|---|---|---|---|---|---|
| 1 | Sandstone | 17.6 | 1.9 | 734.3 | 3.9 | 999.8 | 251.4 | 4.7 | 28.4 |
| 2 | Carbonate | 62.5 | 23.3 | 943.2 | 59.0 | 1162.9 | 666.3 | 37.0 | 57.4 |
| 3 | Carbonate | 53.5 | 20.0 | 859.2 | 53.0 | 1155.0 | 633.2 | 4.6 | 72.1 |
| 4 | Sandstone | 15.2 | 2.9 | 592.1 | 3.6 | 723.3 | 368.5 | 2.7 | 20.8 |
| 5 | Sandstone | 4.0 | 1.0 | 310.2 | 4.0 | 95.0 | 510.1 | 80.3 | 33.5 |
| 6 | Sandstone | 18.4 | 3.3 | 192.9 | 3.3 | 85.1 | 373.4 | 48.9 | 26.2 |
| 7 | Sandstone | 4.8 | 1.0 | 316.9 | 2.9 | 147.5 | 524.7 | 73.5 | 30.0 |
| 8 | Sandstone | 8.0 | 1.5 | 376.7 | 6.0 | 209.9 | 512.5 | 110.0 | 26.3 |
| 9 | Sandstone | 5.6 | 1.0 | 290.5 | 2.1 | 127.6 | 480.8 | 91.6 | 30.2 |
| 10 | Sandstone | 16.0 | 5.3 | 70.6 | 1.9 | 17.0 | 205.0 | 25.3 | 16.4 |
| 11 | Sandstone | 9.6 | 0.1 | 380.8 | 4.5 | 197.1 | 524.7 | 110.3 | 39.3 |
| 12 | Sandstone | 6.4 | 1.0 | 319.7 | 3.7 | 168.8 | 478.4 | 93.7 | 44.2 |
| 13 | Sandstone | 7.2 | 0.5 | 401.0 | 4.9 | 195.7 | 578.4 | 113.1 | 25.6 |
| 14 | Sandstone | 6.4 | 1.9 | 414.0 | 4.4 | 168.8 | 585.8 | 176.8 | 49.3 |
| 15 | Sandstone | 7.2 | 0.7 | 489.6 | 7.4 | 255.6 | 634.2 | 122.4 | 68.0 |
| 16 | Sandstone | 6.7 | 0.7 | 441.7 | 6.0 | 211.8 | 622.2 | 132.8 | 61.1 |
| 17 | Sandstone | 8.2 | 1.1 | 473.4 | 6.4 | 244.2 | 641.7 | 135.6 | 65.8 |
| 18 | Sandstone | 9.6 | 0.5 | 566.7 | 4.1 | 581.4 | 471.1 | 57.4 | 43.4 |
| 19 | Carbonate | 48.0 | 28.7 | 766.9 | 44.6 | 1033.0 | 644.7 | 5.6 | 32.7 |
| 20 | Carbonate | 62.0 | 31.5 | 800.3 | 41.1 | 1033.0 | 618.8 | 24.0 | 41.0 |
| 21 | Carbonate | 66.2 | 33.4 | 855.9 | 51.6 | 1127.0 | 747.7 | 3.0 | 47.1 |
| 22 | Carbonate | 70.1 | 34.4 | 827.4 | 53.6 | 1149.0 | 774.7 | 12.0 | 30.0 |
| 23 | Carbonate | 54.2 | 29.3 | 736.2 | 44.5 | 982.3 | 664.7 | 5.0 | 36.3 |
| 24 | Carbonate | 65.6 | 30.4 | 841.5 | 44.6 | 1084.0 | 685.0 | 40.4 | 46.9 |
| 25 | Carbonate | 67.9 | 34.5 | 786.9 | 38.8 | 989.1 | 683.6 | 63.6 | 40.3 |
| 26 | Carbonate | 49.7 | 25.1 | 872.3 | 43.7 | 1167.0 | 680.3 | 40.2 | 58.1 |
| 27 | Sandstone | 8.3 | 0.8 | 547.1 | 7.1 | 315.1 | 628.2 | 165.1 | 60.9 |
| 28 | Sandstone | 8.2 | 0.6 | 424.3 | 4.6 | 223.9 | 585.2 | 143.5 | 56.3 |
| 29 | Sandstone | 9.6 | 1.9 | 424.2 | 7.1 | 302.1 | 580.9 | 147.8 | 61.3 |
| 30 | Sandstone | 8.2 | 1.1 | 427.9 | 4.7 | 222.8 | 596.8 | 134.0 | 57.9 |
| 31 | Sandstone | 9.1 | 0.9 | 503.8 | 7.1 | 323.9 | 590.3 | 167.9 | 59.9 |
| 32 | Sandstone | 6.4 | 0.7 | 401.7 | 4.6 | 194.3 | 544.0 | 136.6 | 51.0 |
| 33 | Sandstone | 4.0 | 0.6 | 358.1 | 2.3 | 73.6 | 460.7 | 230.9 | 36.7 |
| 34 | Carbonate | 65.7 | 142.5 | 2201.1 | 124.4 | 3034.8 | 341.7 | 760.7 | 62.9 |
| 35 | Carbonate | 255.4 | 45.2 | 1911.0 | 107.1 | 2777.0 | 367.3 | 741.8 | 48.1 |
| 36 | Sandstone | 25.1 | 4.2 | 1138.0 | 12.6 | 1453.0 | 482.1 | 8.0 | 46.0 |
| 37 | Carbonate | 274.3 | 43.1 | 1838.0 | 112.8 | 3000.7 | 371.2 | 487.0 | 49.2 |
| 38 | Carbonate | 221.9 | 46.4 | 1944.0 | 105.1 | 2827.0 | 356.9 | 538.7 | 49.7 |

### 3.3.1. Cation Geothermometer

The cation geothermometer is a geothermometer method established by the relationship between the ratio of Na, K, Ca, Mg and other cations in the geothermal fluid and the temperature. All cation geothermometers are empirical methods [15]. The commonly used cation geothermometers are the Na-K geothermometer, the K-Mg geothermometer and the Na-K-Ca geothermometer. The Na-K geothermometer was first proposed by White (1965) [16]. After that, many foreign scholars proposed similar Na-K geothermometers. This is because a geothermometer may usually only be applied in one area [17,18]. In other areas, there may be larger deviations [19]. The Na-K temperature scale is often used in geothermal water above 150 °C because $Ca^{2+}$ often occupies a certain proportion of the middle- and low-temperature geothermal water [20]. In hot water with higher $Ca^{2+}$ content, the temperature calculated by the Na-K geothermometer is usually higher. In the central-southern area of the Jizhong Depression, although the temperature measured by the boreholes is lower than 100 °C, which is medium–low-temperature geothermal water, the proportion of $Ca^{2+}$ in the geothermal fluid in the study area is relatively low. Therefore, in this paper, the Na-K geothermometer is used to estimate the geothermal

water geothermal reservoir temperature in the study area. This study discusses its practical application in medium- and low-temperature geothermal water in the Jizhong Depression. The geothermal reservoir temperature calculation uses the Na-K geothermometer proposed by Fourner (1979) [21]. The calculation formula is:

$$T = \frac{1217}{\log(Na/K) + 1.483} - 273.15 \tag{1}$$

The K-Mg geothermometer was established by Giggench [18] in 1988. Because K/Mg is more sensitive to temperature changes, it reacts more quickly when the temperature of hot water decreases, thus the K-Mg geothermometer is often used for medium- and low-temperature geothermal water [15]. The calculation formula is:

$$T = \frac{4410}{13.95 - \log(K^2/Mg)} - 273.15 \tag{2}$$

The Na-K-Ca geothermometer is often used in hot water rich in $Ca^{2+}$. Since the Na-K geothermometer cannot be used to obtain a reasonable temperature in hot water rich in $Ca^{2+}$, Fourner and Treusedell [22] based this geothermometer on a lot of practical experience and experiments in 1973. These data develop the Na-K geothermometer into the Na-K-Ca geothermometer, which is often used in medium- and low-temperature geothermal water. The calculation formula is:

$$T = \frac{1647}{\log(Na/Mg) + \beta(\log(\sqrt{Ca}/Na) + 2.06) + 2.47} - 273.15 \tag{3}$$

In the above geothermometer formula: T represents the geothermal reservoir temperature/°C; each ion concentration is mg/l; in the Na-K-Ca geothermometer, $\log(\sqrt{Ca}/Na)$ needs to be calculated first; if $\log(\sqrt{Ca}/Na)$ is positive, then β is taken as 4/3; if $\log(\sqrt{Ca}/Na)$ is negative, then β is taken as 1/3.

3.3.2. SiO$_2$ Geothermometer

SiO$_2$ is commonly found in crustal rocks and minerals such as quartz, chalcedony and clay minerals. SiO$_2$ dissolved in natural water is usually not affected by other ions and is not affected by the formation and volatile dispersion of complexes [23]. As the temperature of hot water decreases, the precipitation process of SiO$_2$ is very slow, especially below 180 °C. The lower the temperature, the longer supersaturation of the SiO$_2$ solution persists [24], thus the relationship between the solubility of SiO$_2$ in hot water and temperature is often used to estimate underground geothermal reservoir temperature. The quartz geothermometer and chalcedony geothermal geothermometer are commonly used to estimate geothermal reservoir temperature. The specific calculation formula is as follows:

Quartz geothermometer without vapor loss:

$$T = \frac{1309}{5.19 - \lg S} - 273.15 \tag{4}$$

Chalcedony geothermometer:

$$T = \frac{1032}{4.69 - \lg S} - 273.15 \tag{5}$$

where T is the temperature (°C) and S is the SiO$_2$ concentration (mg/L).

*3.4. SI Method*

The aforementioned geothermal geothermometers all use the relationship between the chemical composition of geothermal water and temperature to estimate the temperature but fail to judge the chemical equilibrium state between the thermal fluid and minerals in the geothermal system. The Multi-Mineral Balance Graphical Method (SI method) can

use the mineral saturation index (SI) to judge the chemical equilibrium state of minerals and geothermal fluids. When SI > 0, the minerals reach the saturated state; when the SI = 0, the minerals reach the equilibrium state; when SI < 0, the mineral is not saturated. By calculating the mineral saturation index (SI) of the geothermal fluid, the relationship between the dissolved state of different minerals and temperature is analyzed. If a group of minerals is close to equilibrium at a certain temperature, the equilibrium temperature is the geothermal reservoir temperature [25].

*3.5. Water–Rock Balance Analysis*

There are certain assumptions for the use of the geothermal geothermometer. The premise of using the geothermal geothermometer to determine geothermal reservoir temperature is that the ions in the geothermal fluid reach an equilibrium state [26]. Therefore, before using the geothermometer calculation, the water–rock balance state of the geothermal fluid should be judged. The Na-K-Mg triangle diagram is often used to judge the water–rock balance state and whether there is cold water mixing. It divides the geothermal fluid into three types: complete balance, partial balance and immature water [27]. The content of Na, K and Mg in the sample points of the geothermal fluid in the study area is linearly transformed into the Na-K-Mg triangle diagram, and the equilibrium state can be judged according to the position where it falls.

## 4. Results

*4.1. Hydrogeochemistry*

The sandstone geothermal reservoir geothermal fluid in the study area has a pH of 7.78–8.81 with an average value of 8.28, which is weakly alkaline, and a TDS range of 366.5–3193.0 with an average value of 1443.4, which is generally fresh water and brackish water. The pH of carbonate geothermal reservoir geothermal fluid is between 6.90 and 8.87 with an average value of 7.52, which is neutral and weakly alkaline. The TDS ranges from 2639.0 to 6742 with an average value of 3827.2, which belongs to brackish water and saline water. As can be seen in Table 1 and Figure 2, the main ion content of the geothermal fluid in the carbonate geothermal reservoir and the sandstone geothermal reservoir in the study area is different. The content of the main cations $Mg^{2+}$, $Ca^{2+}$, $Na^+$ and $K^+$ in the geothermal fluid of the carbonate geothermal reservoir is higher than that of the sandstone geothermal reservoir. The $Cl^-$ in the carbonate geothermal reservoir anion is higher than the sandstone geothermal reservoir, while $HCO_3^-$ and $SO_4^{2-}$ have some points higher than the sandstone geothermal reservoir and some points lower than the sandstone geothermal reservoir. This is because fluid circulation of the carbonate geothermal reservoir geothermal is deeper than the sandstone geothermal reservoir, and it is generally in a relatively closed environment. The content of ions in the geothermal fluid is related to the runoff length, circulation depth and hydrogeochemical environment of the geothermal fluid [28]. In addition, as can be seen in the Piper three-line diagram (Figure 2), it is found that the cations of the geothermal fluid in the sandstone geothermal reservoir and carbonate geothermal reservoir in the study area are mainly $Na^+$, and the anions are mainly $HCO_3^-$ and $Cl^-$, which are classified in the Shukarev way [29]. The main hydrochemical type of geothermal fluid in the sandstone geothermal reservoir is $HCO_3 \cdot Cl$-Na type water, and the main hydrochemical types of geothermal fluid in the carbonate geothermal reservoir are Cl-Na type water and $Cl \cdot HCO_3$-Na type water.

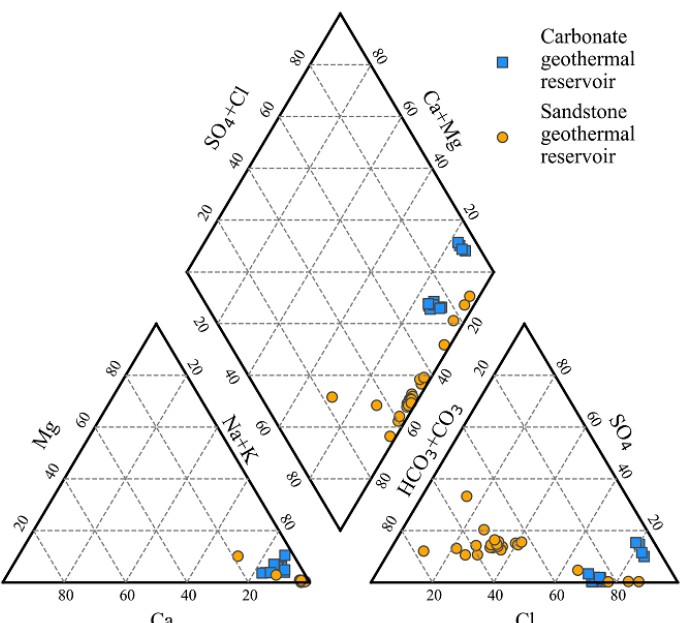

**Figure 2.** Geothermal fluid Piper trilinear diagram.

### 4.2. Borehole Temperature Measurement

The depth of sandstone geothermal reservoir boreholes in the study area was 1510–2425 m, and the geothermal gradient was 2.6–3.0 °C/100 m. The depth of carbonate geothermal reservoir boreholes was 1306–2656 m, and the geothermal gradient was 3.2–3.6 °C/100 m. Based on the borehole logging data, the borehole depth-temperature curve is drawn (see Figure 3). Taking typical boreholes in the study area as an example, points 13 and 16 are sandstone geothermal reservoir boreholes, and points 24 and 34 are carbonate geothermal reservoir boreholes. Whether it is sandstone boreholes or carbonate boreholes, there is a trend of thermal conduction and thermal convection heating. The whole is mainly based on heat conduction, and the heat is transferred by thermal convection in local areas. Heat transfer mainly depends on the rock, and temperature is mainly affected by the thermal conductivity of the rock. In the groundwater active area, the temperature is mainly affected by thermal convection and the geothermal gradient changes greatly. Taking point 13 as an example to calculate the geothermal reservoir temperature, the geothermal reservoir layer ranges from 1900 to 2400 m. The temperature at 1900 m of the top plate is 70.0 °C and the temperature at 2400 m of the bottom plate is 80.0 °C. Taking the average temperature of the top plate and bottom plate of the geothermal reservoir layer as the geothermal reservoir temperature, the temperature of the geothermal reservoir layer can be obtained as 75 °C. All results are shown in Table 2. The borehole ground temperature measurement result is used as the actual temperature for comparison and analysis in relation to other methods.

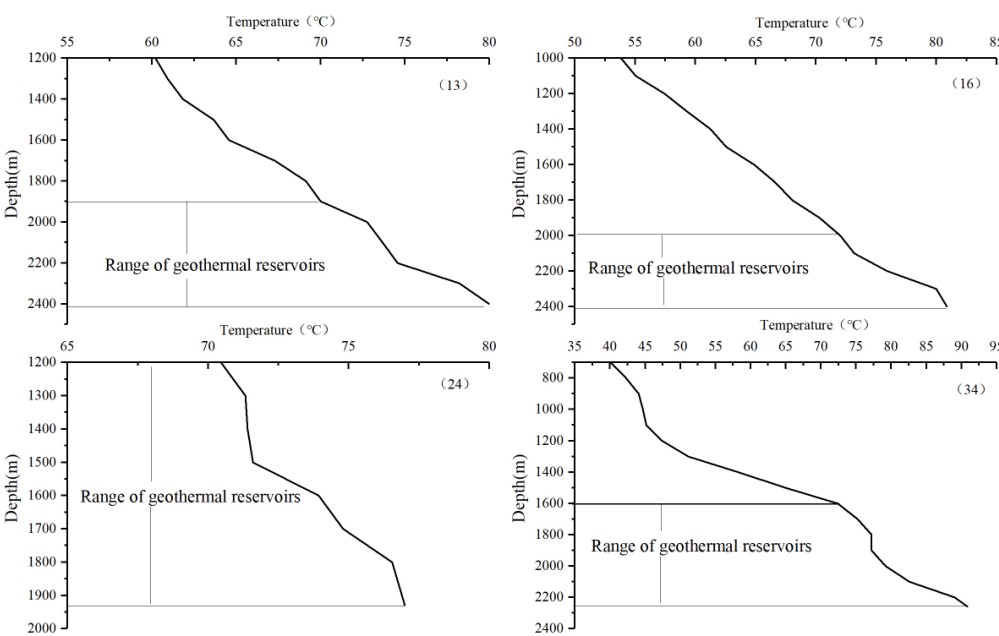

**Figure 3.** Geothermal borehole depth–temperature profile.

*4.3. Geothermal Reservoir Temperature Estimation*

The geothermal reservoir temperature through each geothermometer was calculated, and the results are shown in Table 2. The error value between the estimated and actual temperature is calculated (see Table 3). The error between the estimated value of the Na-K geothermometer and the actual temperature is ±1.3–131.3 °C, the average error is 44.7 °C and the average error percentage is 68%, which greatly overestimates the thermal storage temperature value. The error of the K-Mg geothermometer is ±0.8–42.5 °C, the average error is 15.4 °C and the average error percentage is 24%, which slightly overestimates the thermal storage temperature value. The error of the Na-K-Ca geothermometer is ±9.3–123.3 °C, the average error is 53.7 °C and the average error percentage is 80%, which greatly overestimates the thermal storage temperature value. The error of the quartz geothermometer is ±0.6–42.8 °C, the average error is 22.3 °C and the average error percentage is 32%, which greatly overestimates the thermal storage temperature value. The error of the chalcedony geothermometer is ±0–38.6 °C, the average error is 11.0 °C and the average error percentage is 15%, which slightly underestimates the thermal storage temperature value. It can be seen from the calculation results that, as a whole, the Na-K, Na-K-Ca and quartz geothermometers have large errors, while the errors of K-Mg and chalcedony geothermometers are small.

**Table 2.** Measured temperatures and geothermometer estimates (°C).

| NO. | Measuring Temperature | Na-k | K-Mg | Na-K-Ca | Quartz | Chalcedony | SI Method |
|---|---|---|---|---|---|---|---|
| 1 | 53.0 | 50.7 | 64.9 | 78.5 | 77.1 | 45.7 | 80 |
| 2 | 81.0 | 179.8 | 101.4 | 175.8 | 108.4 | 78.9 | 85 |
| 3 | 98.0 | 178.8 | 100.5 | 174.8 | 119.7 | 91.2 | 82 |
| 4 | 53.0 | 55.8 | 58.4 | 81.4 | 64.9 | 32.9 | 55 |
| 5 | 72.0 | 87.7 | 72.8 | 108.4 | 84 | 52.9 | 65 |
| 6 | 79.0 | 101.3 | 55.2 | 103.5 | 74 | 42.3 | 75 |
| 7 | 80.0 | 72.4 | 65.4 | 94.9 | 79.4 | 48.0 | 90 |
| 8 | 80.0 | 97.8 | 77.7 | 114.7 | 74.1 | 42.5 | / |
| 9 | 68.3 | 62.7 | 58.3 | 84.8 | 79.6 | 48.3 | 70 |
| 10 | 62.4 | 125.5 | 39.2 | 108.9 | 56.1 | 23.7 | 56 |
| 11 | 78.0 | 83.7 | 105.6 | 102.2 | 90.9 | 60.2 | 85 |
| 12 | 80.0 | 82.7 | 71 | 101.8 | 96.1 | 65.8 | 90 |
| 13 | 75.0 | 85.2 | 86.3 | 105.9 | 73 | 41.3 | / |
| 14 | 75.9 | 78.9 | 67.6 | 101.8 | 101.1 | 71.1 | 88 |
| 15 | 88.0 | 95.2 | 92.2 | 116.8 | 116.7 | 88.0 | 94 |
| 16 | 76.0 | 90.0 | 87.3 | 111.5 | 111.4 | 82.2 | 103 |
| 17 | 88.0 | 89.8 | 83.3 | 111 | 115.1 | 86.2 | 85 |
| 18 | 77.6 | 62.7 | 81.8 | 89.4 | 95.3 | 64.9 | 65 |
| 19 | 50.5 | 174.5 | 91 | 170.7 | 83 | 51.8 | 75 |
| 20 | 52.4 | 165.9 | 87.7 | 163 | 92.8 | 62.2 | / |
| 21 | 56.2 | 177.1 | 92.9 | 171.7 | 99 | 68.9 | 75 |
| 22 | 51.0 | 182.3 | 93.5 | 174.3 | 79.5 | 48.1 | 60 |
| 23 | 56.0 | 177.3 | 90.7 | 170.9 | 87.5 | 56.6 | 83 |
| 24 | 73.0 | 168.1 | 90.3 | 164.9 | 98.8 | 68.7 | 80 |
| 25 | 56.0 | 163.1 | 85.1 | 159.9 | 92 | 61.4 | 68 |
| 26 | 80.0 | 164.1 | 92.3 | 165 | 108.9 | 79.6 | 85 |
| 27 | 83.0 | 87.7 | 89.8 | 111.1 | 111.2 | 82.0 | 98 |
| 28 | 78.0 | 80.0 | 82.1 | 101.5 | 107.4 | 77.9 | 102 |
| 29 | 85.2 | 100.2 | 78.9 | 117.1 | 111.6 | 82.4 | 106 |
| 30 | 79.0 | 80.3 | 75.7 | 101.8 | 108.8 | 79.4 | 103 |
| 31 | 81.0 | 92.1 | 88 | 113.1 | 110.4 | 81.1 | 99 |
| 32 | 77.9 | 81.9 | 80.5 | 103.9 | 102.7 | 72.8 | 90 |
| 33 | 75.6 | 57.8 | 66.6 | 84.9 | 88 | 57.1 | 73 |
| 34 | 83.0 | 172.5 | 97 | 185.2 | 112.9 | 83.8 | / |
| 35 | 81.0 | 171.9 | 108.8 | 170.1 | 100 | 70.0 | 89 |
| 36 | 76.4 | 80.8 | 83.2 | 107.5 | 98 | 67.8 | 93 |
| 37 | 75.7 | 178.4 | 111 | 173.6 | 101.1 | 71.1 | 91 |
| 38 | 89.1 | 169.4 | 107.9 | 169.8 | 101.5 | 71.6 | 100 |

Note: The / in the table indicates that no results were obtained using the SI model.

**Table 3.** Error values between estimated and actual temperatures for each geothermometer.

| NO. | Type of Geothermometer | Range of Error (°C) | Average Error (°C) | Average Error Percentage |
|---|---|---|---|---|
| 1 | Na-k | ±1.3–131.3 | +44.7 | 68% |
| 2 | K-Mg | ±0.8–42.5 | +15.4 | 24% |
| 3 | Na-Ca-K | ±9.3–123.3 | +53.7 | 80% |
| 4 | Quartz | ±0.6–42.8 | +22.3 | 32% |
| 5 | Chalcedony | ±0–38.6 | −11.0 | 15% |
| 6 | SI method | ±1.7–27.0 | +12.6 | 18% |
| 7 | Points on the Na/K line | ±1.3–22.3 | +8.4 | 11% |
| 8 | Points on the chalcedony line | ±0–12.7 | −3.0 | 5% |
| 9 | Points on the quartz line | ±0.6–42.8 | +23.9 | 34% |
| 10 | K-Mg (in sandstone) | ±0.8–27.6 | +9.1 | 11% |
| 11 | K-Mg (in carbonate) | ±2.5–42.5 | +26.4 | 40% |
| 12 | NA-K-Ca (in sandstone) | ±9.3–46.5 | +26.4 | 34% |
| 13 | NA-K-Ca (in carbonate) | ±76.8–123.3 | +100.5 | 141% |

Note: In the average error, the + sign indicates high estimates and the − sign indicates low estimates.

### 4.4. SI Method

Since the water quality analysis did not obtain the exact aluminum content, this paper uses the fixed aluminum value method [30] to calculate the mineral saturation index. Select the common minerals (quartz, chalcedony, illite, calcium montmorillonite, albite, etc.) in the study area [31] and use Pheeqc software [25] to simulate the saturation index of each mineral within 50–200 °C. Take temperature as the abscissa and the saturation index SI as the ordinate to make the SI-T diagram in the equilibrium state.

It can be seen from Figure 4 (only some points are listed) that most of the minerals have intersections with SI = 0; however, most of the points only have two minerals that intersect at SI = 0. The temperature at which the two minerals intersect at SI = 0 is used as the geothermal reservoir temperature. Except for a few points where no minerals intersect at SI = 0 and no result is obtained, all other points have obtained results. The results are shown in Table 2 (note: in the table / indicates unobtained results). Comparing the obtained results with the actual temperature, the error is ± 1.7–27.0 °C, the average error is 12.6 °C and the average error percentage is 18%, slightly overestimating the thermal storage temperature value.

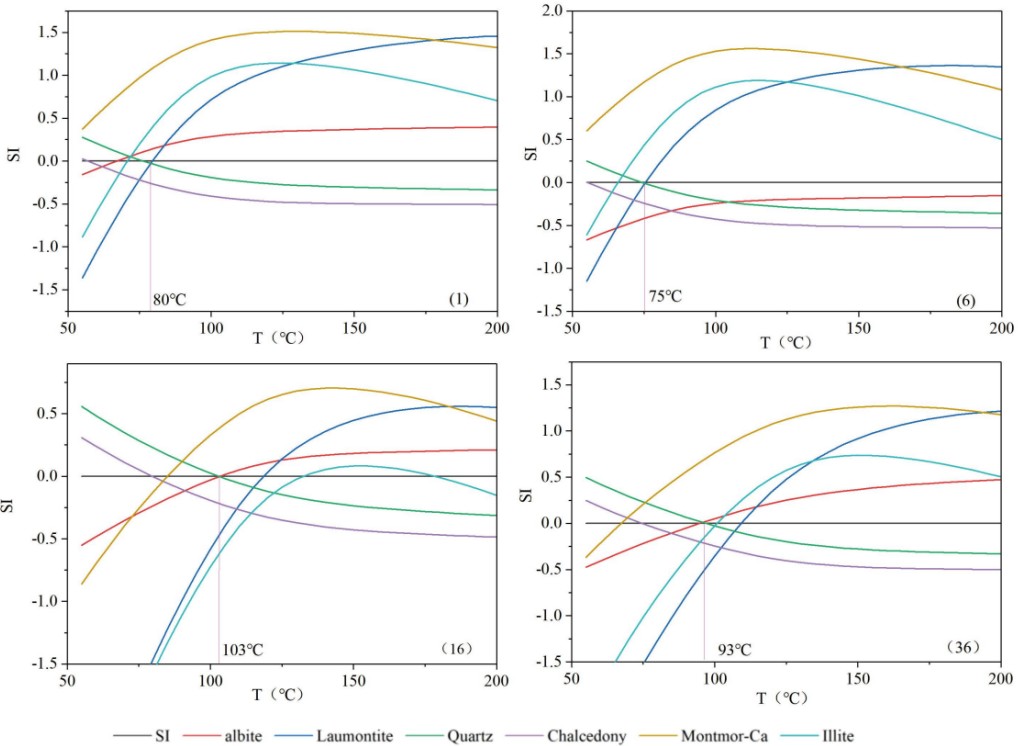

**Figure 4.** SI-T diagram for geothermal fluids (note: vertical lines are multiple minerals with SI values at this temperature intersecting at SI = 0, using this temperature as the geothermal reservoir temperature).

### 4.5. Water–Rock Balance Analysis

The Na-K-Mg triangle diagram of the water sample points in the study area was drawn using Auqchem software (Figure 5). It can be seen from Figure 5 that the geothermal water in the study area mainly falls in the immature water area and part of the equilibrium area. Falling in the immature water area indicates that it is in the initial stage of water–rock balance, while falling in the partial balance area reflects that the geothermal fluid has not reached the ion balance state and the dissolution is still proceeding or being mixed by cold water. Generally speaking, points 2, 3, 6, 10, 19, 20, 21, 22, 23, 24, 25, 26 and 34, located in the immature water area, do not apply the cation geothermometer to calculate

the geothermal reservoir temperature and fall in the part where the balance zone needs to further determine whether the cationic geothermometer is applicable.

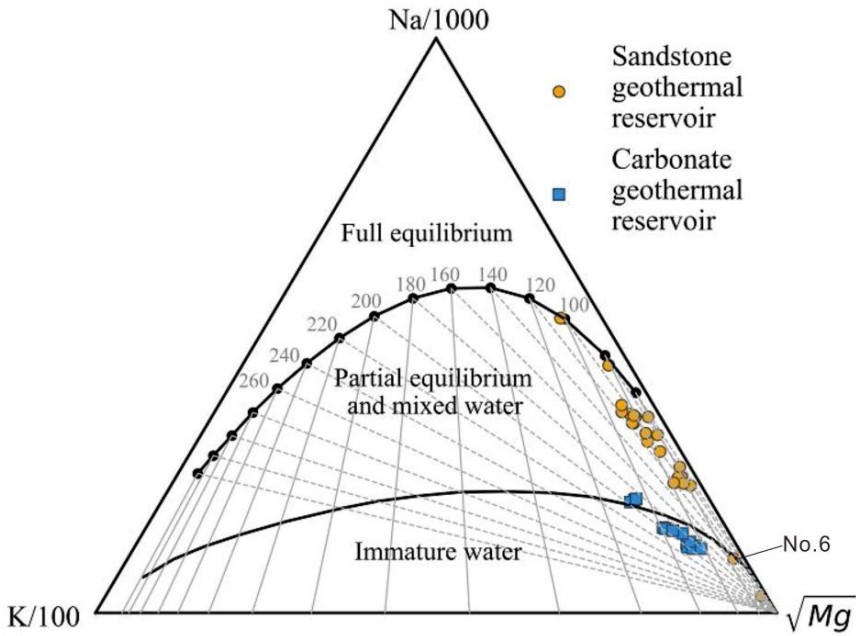

**Figure 5.** Diagrammatic representation of the Na-K-Mg equilibrium of a geothermal fluid.

In order to further analyze the applicable geothermometer of the water samples in the study area, Auqchem software [32] was used to draw the relationship between $SiO_2$ and 1000/T in the study area (Figure 6a) and the relationship between Na/k and 1000/T (Figure 6b). From Figure 6a, it can be found that most of the water samples in the study area (except for points 6, 8 and 13) are above the theoretical equilibrium line of quartz, indicating that these points can be calculated with quartz geothermometers. For points that intersect the chalcedony theoretical balance line or points above it (i.e., 2, 15, 16, 17, 19, 20, 21, 22, 23, 24, 25, 26, 27, 28, 29, 30 and 31), the chalcedony geothermometer can be used for temperature calculation. In Figure 6b, points 1, 4, 5, 6, 7, 8, 9, 11, 12, 13, 14, 15, 16, 17, 18, 27, 28, 29, 31, 32, 33 and 36 can be seen above the theoretical balance line of Na/K, showing that these points can be calculated with the Na-K geothermometer. It can also be found that these points are all sandstone geothermal reservoir points. Among the 24 sandstone geothermal reservoir points, only one point is located below the Na/K theoretical balance line, indicating that Na/K is suitable for most sandstone geothermal reservoir points in the study area, while carbonate geothermal reservoir points are all located below the Na/K theoretical equilibrium line, indicating that the carbonate geothermal reservoir points are not applicable to the Na/K geothermometer. In addition, from Figure 6b, it can be found that all points in the study area are located below the Na-K-Ca theoretical equilibrium line, indicating that the geothermal fluid in the study area is not suitable for calculation using the Na-K-Ca geothermometer. As can be seen in the comprehensive Na-K-Mg diagram and Na/k and 1000/T relationship diagram, it is found that although point 6 is located in the immature water area, it is still on the theoretical balance line of Na/K, indicating that point 6 may be greatly affected by mixing. At the same time, some scholars have proposed that the mixing effect of cold and heat has little effect on the Na/K value [27].

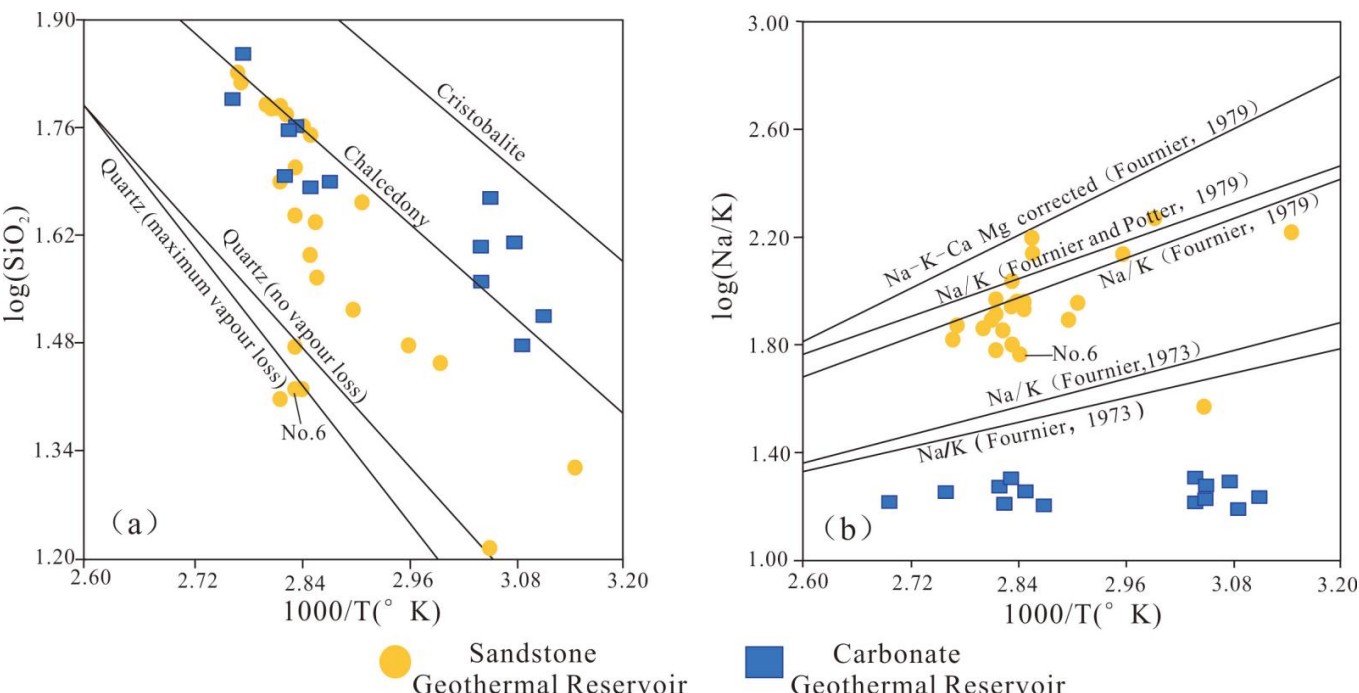

**Figure 6.** Diagram of (**a**) SiO2 and (**b**) Na/K at 1000/T for geothermal fluids.

## 5. Discussion

By using each geothermal geothermometer to calculate the geothermal reservoir temperature of the sample point, it is found that the same sample point uses different geothermometers, and the results of the geothermal reservoir temperatures obtained are also different. From the temperature comparison chart (Figure 7), it can be seen that the estimated value of the multi-mineral balance method is the closest to the measured temperature on the whole and the fluctuation is the smallest. Most of the estimated values of K-Mg and chalcedony geothermometers are closer to the actual measured temperature but there are some points that fluctuate greatly. The estimated value of the Na-K geothermometer also has some points that are closer to the actual measured temperature but there are many points that fluctuate greatly. There is a significant gap between the estimated value curve of the quartz geothermometer and the actual measured value curve. The largest overall difference is the estimated value of the Na-K-Ca geothermometer. On the whole, based on the results of the error values (Table 3) and Figure 7, the SI method and K-Mg and chalcedony geothermometers have relatively small errors and are more suitable for the central and southern Jizhong Depression. However, to be practically applied to geothermal development, there are still some points with large errors. It is necessary to further analyze the conditions of each geothermometer to find appropriate methods to reduce the differences and provide a more reliable reference basis for geothermal development.

In the previous chapter, water–rock balance analysis was performed on the geothermal fluid and the Na-K-Mg balance diagram, the relationship diagram between $SiO_2$ and 1000/T and the relationship diagram between Na/K and 1000/T were drawn. Geothermal fluid was then applied and the geothermometer was analyzed. According to the analysis results, the points located above the theoretical balance line of Na/K are analyzed, and it is found that the points located on the theoretical balance line of Na/K have a small error between the temperature estimated by the Na-K geothermometer and the actual measured temperature. The error is ±1.3–22.3 °C. The average value is 8.4 °C, which is significantly smaller than the error without condition analysis and judgment. The larger error values are all carbonate geothermal reservoir points, indicating that the carbonate geothermal reservoir in the central part of the Jizhong Depression is not suitable for calculations using

the Na-K geothermometer, though the Na-K geothermometer is suitable for most sandstone geothermal reservoir points. In addition, all points in the study area are located below the Na-K-Ca theoretical equilibrium line and the Na-K-Ca geothermometer is not suitable for the calculation of storage temperature in the study area. Consistent with the calculated results, the estimated value of the Na-K-Ca geothermometer has the largest error in relation to the actual temperature.

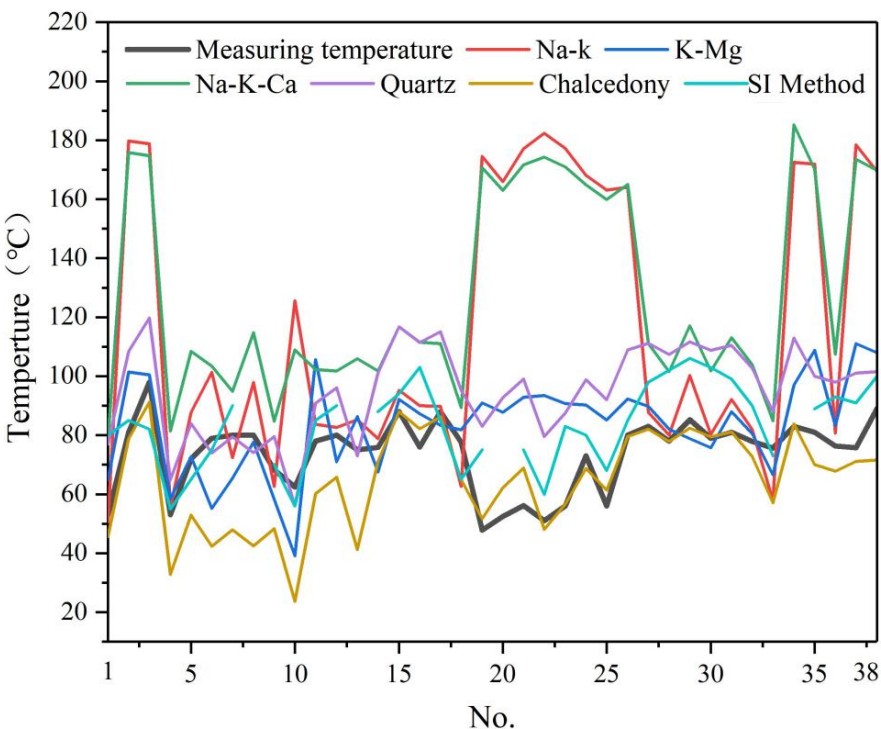

**Figure 7.** Comparison of measured and estimated temperatures.

It can be seen from the analysis results of the Na-K geothermometer that the smaller errors between the estimated temperature and the actual measured temperature of the Na-K geothermometer are all sandstone geothermal reservoir samples. Combining the calculation results, it can be found that the same is true for the K-Mg geothermometer and the Na-K-Mg geothermometer. The error range of the K-Mg geothermometer in carbonate geothermal reservoir samples is ±2.5–42.5 °C, and the average error is 26.4 °C. In the sandstone geothermal reservoir, the error range is ±0.8–27.6 °C and the average error is 9.1 °C. The error value of the Na-K-Ca geothermometer in the carbonate thermal water storage sample is ±76.8–123.3 °C and the average error is 100.5 °C. The error value in the sandstone geothermal reservoir water sample is ±9.3–46.5 °C and the average error is 26.4 °C.

Although the Na-K geothermometer is often used in geothermal water at 150 °C, in the central area of the Jizhong Depression, the average error of the points on the Na-K line after water–rock balance analysis is only 8.4°C. Previous studies have shown that the error of the cation geothermometer under the most suitable conditions will be ±5–10 °C; however, under normal circumstances, the error is far more than ±20 °C [15], indicating that the Na-K geothermometer can be used in medium- and low-temperature geothermal water systems. On the whole, the Na-K-Ca geothermometer of the cationic geothermometer is not applicable to the central-southern area of the Jizhong Depression. The Na-K geothermometer and K-Mg geothermometer are applicable to most of the sandstone geothermal reservoirs in the central-southern Jizhong Depression, and they are less suited to use in carbonate geothermal reservoirs.

The error of the points above the theoretical balance line of quartz is ±0.6–42.8 °C and the average error is 23.9 °C. The overall difference has not been reduced compared to the previous one. Some scholars have shown that the quartz geothermometer represents the highest temperature reached by the geothermal reservoir [33,34]. The error between the point on the chalcedony line and the measured temperature is ±0–12.7 °C with an average error of 3.0 °C, and the overall error is significantly reduced. According to previous research results, the geothermometer of chalcedony is most suitable for medium- and low-temperature geothermal water, and quartz is suitable for geothermal water at high temperature [35]. Consistent with this, the Jizhong Depression is all low-temperature geothermal water, and the overall error of the chalcedony geothermometer is the smallest. It is most suitable for the central and southern areas of the Jizhong Depression, while the quartz geothermometer is not applicable to the central and southern areas of the Jizhong Depression.

A box-type comparison chart is made between the direct use of the geothermal geothermometer to calculate the geothermal reservoir temperature (Figure 8a) and the selective use of the geothermal geothermometer after water–rock balance analysis (Figure 8b). It can be seen from the comparison chart that the selective use of geothermal temperature scale after water–rock balance analysis is better than direct use, and the error is significantly reduced. Chalcedony, Na-K and K-Mg geothermometers and multi-mineral balance methods have achieved better results. The geothermometer error between Na-K-Ca and quartz is still relatively large. The geothermal reservoir temperature determines the development and utilization value of geothermal resources, but the geothermal geothermometer is an empirical formula based on a large amount of practical experience and experiments obtained by predecessors and its use conditions are limited. Therefore, when estimating geothermal reservoir temperature, it cannot be used directly, which will cause big differences and incorrectly estimate the geothermal reservoir temperature. Water–rock balance analysis should be performed, and the applicable geothermometer of geothermal fluid can then be determined by using the Na-K-Mg balance diagram, the relationship diagram between $SiO_2$ and 1000/T and the relationship diagram between Na/K and 1000/T. Use after judgment, so as to provide a reliable basis for the development and utilization of geothermal resources.

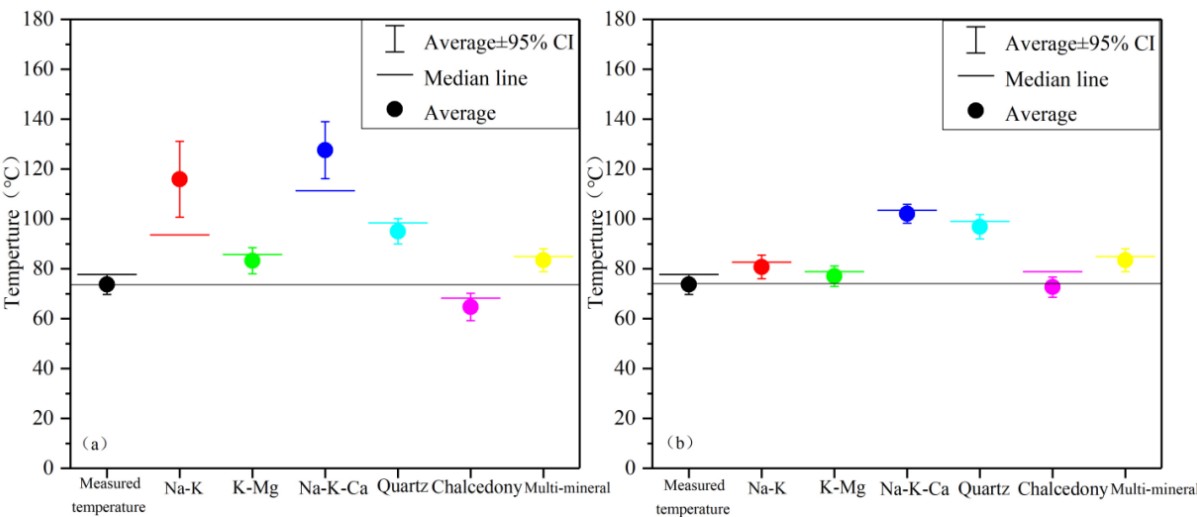

**Figure 8.** Comparison of direct use (**a**) and selective use (**b**) of geothermometer box types.

## 6. Conclusions and Suggestions

(1) After analyzing 38 geothermal fluid sample points in the central and southern Jizhong Depression, it can be found that the hydrochemical type of the sandstone geothermal reservoir geothermal fluid is mainly $HCO_3 \cdot Cl$-Na type water, and the carbonate geothermal reservoir geothermal fluid is mainly Cl-Na type water and $Cl \cdot HCO_3$-Na type

water. The cation geothermometer, silica geothermometer and multi-mineral balance method were used to estimate geothermal reservoir temperature and compare it to the actual temperature obtained by borehole ground temperature measurement. Through comparison, it can be found that the direct use of the geothermal geothermometer will cause large errors. The water–rock balance analysis should be carried out first, and the Na-K-Mg balance diagram, the relationship diagram between $SiO_2$ and $1000/T$ and the relationship diagram between Na/K and $1000/T$ should be used to analyze and judge. After analysis and judgment, the use of geothermal geothermometers can effectively reduce the error.

(2) After research and analysis to determine the applicable geothermometer for geothermal fluids, in the cationic geothermometer, except for the poor use of Na-K-Ca geothermometers, both Na-K and K-Mg geothermometers have achieved good results and they are effective in the central and southern Jizhong Depression. Most of the sandstone geothermal reservoir in the area is applicable, though the use error of the carbonate geothermal reservoir is relatively large. The SI method has also achieved good results and can be applied to the central and southern Jizhong Depression. The quartz geothermometer has a poor effect and is not suitable for the central and southern area of the Jizhong Depression. The chalcedony geothermometer had the best effect and is the most applicable geothermometer in the central and southern Jizhong Depression.

(3) When using a geothermal geothermometer, it should be used after research and analysis, comprehensively comparing the results of each geothermometer and selecting the appropriate geothermal reservoir temperature in order to provide a reliable basis for the development and utilization of geothermal energy.

**Author Contributions:** Conceptualization, J.D. and W.L.; methodology, W.L.; software, W.L.; validation, J.D.; formal analysis, W.L. and L.C.; investigation, J.D. and L.X.; data curation, L.X.; Project administration, W.L.; writing—original draft preparation, J.D.; writing—review and editing, J.D., W.L. and L.X.; visualization, W.L. and L.C.; supervision, L.C.; funding acquisition, W.L. All authors have read and agreed to the published version of the manuscript.

**Funding:** This research was supported by the National Key R&D Program of China (Grant No. 2021YFB1507401) and the China Geological Survey (Grant No. DD20190555).

**Institutional Review Board Statement:** Not applicable.

**Informed Consent Statement:** Not applicable.

**Data Availability Statement:** The data used to support the findings of this study are included within the article.

**Conflicts of Interest:** The authors declare no conflict of interest.

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
