# Peer review of "The Estimation of Geothermal Reservoir Temperature Based on Integrated Multicomponent Geothermometry: A Case Study in the Jizhong Depression, North China Plain"

_water, doi:10.3390/w14162489_

Round 1
Reviewer 1 Report
General Remarks:
This is an interesting study to compare the measured bottom hole temperatures with different chemical geothermometers in the Jizhong depression. Unfortunately, comparison with results of international modern geothermometry studies is missing. In addition, in some parts the English is difficult to read, reference numbering is chaotic and obviously problems with the spacing at the end of sentences exist.
Specific Remarks:
p.2 paragraph 2 l.5: …heat storage...
Figure 1b: Add numbers or range of numbers of the geothermal drill holes to better understand the regional distribution of geothermal temperatures. In this connection a schematic cross-section would be even more informative.
p.3 p. 2 l. 11l. Give the standard error of the analysed cat-, anions, and SiO2. Describe how SiO2 were analysed.
p. 4 p. 1 l. 2: Spell full name of abbreviations mentioning the first time.
p. 4 p. 1 l. 3: Mention here or in the discussion about the problematic to measure the true formation temperature shortly after finishing a drill hole. In addition, the bottom of the drill-hole may not the place of hydrochemical equilibration.
p. 5.p. 1 l. 5 Reference of White (1965) is missing.
p. 5.p. 1 l. 11 This is not an English sentence and has a capital letter in the middle of the sentence.
p. 5.p. 1 l. 6 Cite at least some of these authors.
p. 5.p. 1 l. 18 Reference number of Fourner (1979) is missing here.
Formula (1): Why in this formula k is written as lower-case character?
p. 5.p. 2 l. 1 Reference number of Giggench (1988) is missing here.
p. 5.p. 3 l. 3 Reference number of Fourner and Treusedell (1973) is missing here
p. 7.p. 1 l. 11 Cite the paper of the Shukarev classification.
Figure 2 This figure is not essential and gives not more information than Table 1 and Figure 3.
p. 8.p. 1 l. 8 Report the range of temperature gradients of the sandstone and carbonate geothermal drill-holes.
Figure 4 Show the full range of ground temperature measurements from the surface to the bottom of the drill-holes. The groundwater advection would be still visible.
p. 8.p. 2 l. 4 Generally a hyphen symbol is used between the range numbers. Is there a special argument to use “approximate symbol” ~ (circumflex)?
p. 8.p. 2 l. 4 Indicate if the average error is above or below the bottom temperature measurement in the text and in Table 3. It is important to see immediately if the calculations of the geothermometers over- or underestimate the temperature.
p. 8.p. 3 l. 4 Cite Pheeqc or Phreeqc software.
p. 9.p. 1 l. 5 Explain in more detail: “Results are obtained at other points”.
p. 9.p. 2 l. 2 Cite Auqchem software and who has introduced this diagram into the scientific literature.
p. 12 p. 1 l. 2 Point out No. 6 in the diagrams.
Figure 8: To enhance the visibility of the “measured temperature” increase the thickness of the black line. Instead of NO. on the x-axis write more common No.
p. 14 p. 1 l. 1 …points?...
References
[16] Add publisher and the city of the publishing house.
Author Response
Dear Editors and Reviewers:
Thank you for your letter and for the reviewers’ comments concerning our manuscript entitled “The Estimation of Geothermal Reservoir Temperature Based on the Integrated Multicomponent Geothermometry: A Case Study in Jizhong Depresion, North China Plain” (ID: water-1837065). Those comments are all valuable and very helpful for revising and improving our paper, as well as the important guiding significance to our researches. We have studied comments carefully and have made correction which we hope meet with approval. Revised portion are marked in red in the paper. The main corrections in the paper and the responds to the reviewer’s comments are as flowing:
Responds to the reviewer’s comments:
- 2 paragraph 2 l.5: …heat storage...
Response:Corrected in paper.
- Figure 1b: Add numbers or range of numbers of the geothermal drill holes to better understand the regional distribution of geothermal temperatures. In this connection a schematic cross-section would be even more informative.
Response:A sample number or range has been added to the plot.
- 3 p. 2 l. 11l. Give the standard error of the analysed cat-, anions, and SiO2. Describe how SiO2 were analysed.
Response:A description has been added to the paper.
- 4 p. 1 l. 2: Spell full name of abbreviations mentioning the first time.
Response:Changed to full name in paper.
- 4 p. 1 l. 3: Mention here or in the discussion about the problematic to measure the true formation temperature shortly after finishing a drill hole. In addition, the bottom of the drill-hole may not the place of hydrochemical equilibration.
Response:Discussion of this issue has been added to the paper.
- 5.p. 1 l. 5 Reference of White (1965) is missing.
Response:References have been added to the paper
- 5.p. 1 l. 11 This is not an English sentence and has a capital letter in the middle of the sentence.
Response:The sentence has been reworked in the paper
- 5.p. 1 l. 6 Cite at least some of these authors.
Response:References have been added to the paper.
- 5.p. 1 l. 18 Reference number of Fourner (1979) is missing here.
Response:References have been added to the paper.
- Formula (1): Why in this formula k is written as lower-case character?
Response:We are very sorry to have overlooked this issue, which has been corrected in the paper.
- 5.p. 2 l. 1 Reference number of Giggench (1988) is missing here.
- 5.p. 3 l. 3 Reference number of Fourner and Treusedell (1973) is missing here
- 7.p. 1 l. 11 Cite the paper of the Shukarev classification.
Response:We have made corrections based on reviewer comments and added references to the paper.
- Figure 2 This figure is not essential and gives not more information than Table 1 and Figure 3.
Response:As the reviewer stated, Figure 2 appears redundant and has been removed from the paper.
- 8.p. 1 l. 8 Report the range of temperature gradients of the sandstone and carbonate geothermal drill-holes.
Response:Based on your comments, the scope of the geothermal gradient has been increased in the paper.
- Figure 4 Show the full range of ground temperature measurements from the surface to the bottom of the drill-holes. The groundwater advection would be still visible.
Response:Re-described in the paper.
- 16 . 8.p. 2 l. 4 Generally a hyphen symbol is used between the range numbers. Is there a special argument to use “approximate symbol” ~ (circumflex)?
Response:We're sorry to have overlooked this issue and have corrected it in the paper.
- 8.p. 2 l. 4 Indicate if the average error is above or below the bottom temperature measurement in the text and in Table 3. It is important to see immediately if the calculations of the geothermometers over- or underestimate the temperature.
Response:A description has been added for this issue in Table 3 and the paper.
- 8.p. 3 l. 4 Cite Pheeqc or Phreeqc software.
Response:Added citations to relevant references.
- 9.p. 1 l. 5 Explain in more detail: “Results are obtained at other points”.
Response:We're sorry for the erroneous description, which has been corrected in the paper.
- 9.p. 2 l. 2 Cite Auqchem software and who has introduced this diagram into the scientific literature.
Response:Added citations to relevant references.
- 12 p. 1 l. 2 Point out No. 6 in the diagrams.
Response:Added the location of point 6 to Figures 5 and 6.
- Figure 8: To enhance the visibility of the “measured temperature” increase the thickness of the black line. Instead of NO. on the x-axis write more common No.
Response:Figure 8 has been modified according to your suggestion.
- 14 p. 1 l. 1 …points?...
Response:Modified in the paper.
- [16] Add publisher and the city of the publishing house.
Response:Added the publisher's city to the reference at your suggestion.
We tried our best to improve the manuscript and made some changes in the manuscript. These changes will not influence the content and framework of the paper. All modifications are marked in red in the paper.
Once again, thank you very much for your comments and suggestions.

Reviewer 2 Report
The article is interesting and presents a comparison of field samples with different temperature estimation methods. It may be necessary to introduce more information on sample collection. Also several sentences are not clearly written and a revision of the article in detail on language clarity is recommended. Further comments and suggestions for improvement can be found in the attached document.

Author Response
Dear Editors and Reviewers:
Thank you for your letter and for the reviewers’ comments concerning our manuscript entitled “The Estimation of Geothermal Reservoir Temperature Based on the Integrated Multicomponent Geothermometry: A Case Study in Jizhong Depresion, North China Plain” (ID: water-1837065). Those comments are all valuable and very helpful for revising and improving our paper, as well as the important guiding significance to our researches. We have studied comments carefully and have made correction which we hope meet with approval. Revised portion are marked in red in the paper. The main corrections in the paper and the responds to the reviewer’s comments are as flowing:
Responds to the reviewer’s comments:
1.What the authors means with "coal-fired energy structure"?
Response:It has been modified in the text to coal-dominated energy structure, which means the energy structure with a high proportion of coal.
2.Avoid repetition
Response:Sentences have been revised in the article.
3.This phrase is not clear, please, rewrite it
Response:Several of these questions in the article have been rewritten at your request.
4.What is a genetic type?
Response:Incorrect wording, changed to genesis type.
5.Please, explain better how the authors get the fluid samples. They are from open wells? or they had to drill to get them? at which depth they were taken? etc..
Response:According to your request, the relevant description has been added to the article.
6.From here, a explanation of different methods is written. but you have to introduce to the reader that you are going to do this. Then, a previous paragrath explain the structure is needed.
Response:As suggested by the reviewer, we have added the relevant description in the previous section.
7.To avoid missunderstanding, this paragraght will be better 3.3.1
Response:According to your suggestion, it has been changed to 3.3.1 in the text.
8.No clear. "the borehole depth.temperature curve diagram is drawn" is better.
Response:This sentence has been revised in the text according to your suggestion.
9.From here, maybe it would be interesting to add a column with the error in % in all the followings tables with error informatiton.
Response:As you suggested, adding the error percentage makes the result clearer. However, due to the limitation of the framework of Table 2, we added the average error percentage in Table 3.
We tried our best to improve the manuscript and made some changes in the manuscript. These changes will not influence the content and framework of the paper. All modifications are marked in red in the paper.
Once again, thank you very much for your comments and suggestions.
